# Equilibrium selection via current sheet relaxation and guide field amplification

Young Dae Yoon [1,2] ✉, Deirdre E. Wendel [3] & Gunsu S. Yun [4,5] ✉

Although there is a continuous spectrum of current sheet equilibria, how a particular equilibrium is selected by a given system remains a mystery. Yet, only a limited number of equilibrium solutions are used for analyses of magnetized plasma phenomena. Here we present the exact process of equilibrium selection, by analyzing the relaxation process of a disequilibrated current sheet under a finite guide field. It is shown via phase-space analyses and particle-in-cell simulations that the current sheet relaxes in such a way that the guide field is locally amplified, yielding a mixed equilibrium from the spectrum. Comparisons to spacecraft observations and solar wind current sheet statistics demonstrate that such mixed equilibria are ubiquitous and exist as underlying local structures in various physical environments.

Current sheets are structures that are ubiquitous in space and laboratory plasmas[1–6]. Sandwiched by two regions of opposing magnetic fields, current sheets act as magnetic batteries which store magnetic free energy that can be converted to other forms of energy. Examples of such conversion processes include important phenomena such as magnetic reconnection[7,8], drift kink instability[9], ideal tearing instability[10], and plasmoid instability[11,12]. Current sheets have thus been subject to extensive research over the past few decades.

A subject that has accrued much interest in particular is current sheet equilibria, which determine the stationary profiles of various plasma parameters such as magnetic field and temperature. A plethora of analytical and numerical equilibria have been found[13–24], and these are readily used as initial conditions for analyses of various plasma phenomena. For instance, two current sheet equilibrium solutions—the Harris equilibrium[13] and the force-free equilibrium[22]—are notably and almost exclusively used for magnetic reconnection analyses[25–28].

A relatively less explored subject is how initially disequilibrated current sheets relax to equilibrium states. Such knowledge is important because plasma systems in general start from disequilibrated states and so it reveals how such systems tend to dynamically evolve in time. Also, the relaxation process manifests the general solution of current sheet equilibria, providing a more profound insight than the multitude of specific solutions that have been found so far. For instance, Yoon et al.[29] showed that current sheet equilibration—heating and pinching—is achieved via particle orbit class transitions, or, in other words, that a particular equilibrium solution can be represented by the fractional population in each orbit class. A useful by-product of this analysis was the prediction of bifurcated current sheets[30], whose origins had remained controversial for over a decade. The ubiquity of bifurcated structures was explained by showing that they are natural outcomes of the relaxation process of disequilibrated states, from which current sheets most likely start their dynamics.

A crucial limitation to the analysis in Yoon et al.[29] was that it did not include effects due to the magnetic field component parallel to the current direction, i.e., the guide field. Guide field effects, however, are of paramount importance because zero-guide-field current sheets rarely exist in reality, and the presence of even the smallest guide fields significantly alter ensuing dynamics, e.g., in guide field reconnection[31,32]. In fact, spacecraft observations seem to indicate that finite guide fields deter the formation of bifurcated current sheets[33–37]; the origin of this dissimilarity with the zero-guide-field case is unclear. As such, current sheet relaxation under guide fields is a different matter to deal with and must therefore be thoroughly investigated in comparison to the zero-guide-field case.

Finite guide fields are also important in regards to equilibrium selection. Although the two aforementioned equilibrium solutions—Harris and force-free—are usually employed for many practical purposes, there is actually a continuous spectrum of solutions which can

[1]Asia Pacific Center for Theoretical Physics, Pohang, Gyeongbuk 37673, Republic of Korea. [2]Pohang Accelerator Laboratory, POSTECH, Pohang, Gyeongbuk 37673, Republic of Korea. [3]NASA Goddard Space Flight Center, Greenbelt, MD 20771, USA. [4]Department of Physics and Division of Advanced Nuclear Engineering, POSTECH, Pohang, Gyeongbuk 37673, Republic of Korea. [5]Center for Attosecond Science, Max Planck POSTECH/Korea Research Initiative, Pohang, Gyeongbuk 37673, Republic of Korea. ✉e-mail: youngdae.yoon@apctp.org; gunsu@postech.ac.kr

be represented by combinations of the two equilibria, which are respectively at each end of the spectrum[22,38–41]. The frequent usage of only the two extremes is therefore not justified. It is also not clear how a particular equilibrium out of the spectrum is selected or generated by a given system, e.g., how a zero-guide-field Harris sheet transitions to a sheet with a locally amplified guide field such as the force-free sheet.

In fact, statistical studies of current sheets in various space enviroments have shown that combinations of the two equilibria (hereafter dubbed "mixed equilibria") are prevalent[42–46]. For example, Panov et al.[43] showed that out of the 52 observed current sheets in the magnetopause, around half of them are "C-shaped," which corresponds to the mixed equilibria as per our definition. Artemyev et al.[44] also showed that 123 out of 226 observed current sheets in the Jovian magnetotail are "type 2," which are our mixed equilibria. Lotekar et al.[45] and Vasko et al.[46] each studied the statistics of >10,000 current sheets in the near-Earth and near-Sun solar winds, respectively. They showed that the average current sheet is nearly force-free with a slight local dip in the magnetic field strength, which is also a mixed equilibrium.

Here we present, via an analysis of the collisionless relaxation mechanism of a current sheet under a finite guide field, the process in which a particular equilibrium is selected by a given plasma system. This is shown in the following steps; first, particle orbits in a magnetic field reversal under a finite guide field are classified into three distinct orbit classes. The guide field induces an anti-symmetry in the particle phase space distribution. Second, it is then shown via comparisons to particle-in-cell (PIC) simulations that a disequilibrated (under-heated) Harris sheet with a small seed guide field relaxes via orbit class transitions. Ions are responsible for sheet heating and electrons are responsible for sheet pinching and shearing. The transitions are in such a way that the seed guide field is locally amplified at the center, yielding one of the equilibria within the spectrum, i.e., a mixture of Harris and force-free equilibria. The distribution anti-symmetry is responsible for guide field amplification, but the phenomenon can also be intuitively understood in the magneto-hydrodynamic (MHD) picture. The reduction of bifurcation under finite guide fields is naturally explained by the type of orbit class transitions that take place. Finally, the equilibrium current sheet from the PIC simulation is compared with a current sheet observed by the Magnetospheric Multiscale (MMS) mission, thereby confirming the theoretical and numerical analyses. The bearing of these results on solar wind current sheet statistics explains the origin and universality of mixed equilibria.

## Results
### Equilibrium spectrum
The spectrum of one-dimensional equilibria represented by a mixture of Harris and force-free current sheets can be expressed by the electrostatic potential $\phi = 0$ and the following values of the magnetic field **B**, the plasma pressure tensor component $P_{xx}$, and the current density **J** in Cartesian coordinates[22]:

$$\mathbf{B} = B_0 \left[ \hat{y} \tanh\left(\frac{x}{\lambda}\right) + \hat{z}\frac{b_g}{\cosh(x/\lambda)} \right], \quad (1)$$

$$P_{xx} = \frac{B_0^2}{2\mu_0}\left(1 - b_g^2\right)\frac{1}{\cosh^2(x/\lambda)}, \quad (2)$$

$$\mathbf{J} = \frac{B_0}{\mu_0\lambda}\left[ \hat{y}b_g\frac{\sinh(x/\lambda)}{\cosh^2(x/\lambda)} + \hat{z}\frac{1}{\cosh^2(x/\lambda)} \right], \quad (3)$$

where $B_0$ is the asymptotic magnetic field strength, $b_g$ is the guide field strength relative to $B_0$, and $\lambda$ is the sheet half-thickness. Note that the

system being studied is 1D in configuration space and 3D in velocity space. $x$ is normal to the current sheet, $y$ is along the shear magnetic field, and $z$ is along the guide field. Taking $b_g = 0$ yields the Harris equilibrium where the asymptotic magnetic pressure $B_0^2/2\mu_0$ is balanced by the thermal pressure at the center ($x = 0$), and taking $b_g = 1$ yields the force-free equilibrium where $|\mathbf{B}| = B_0$ everywhere and so $\mathbf{J} \times \mathbf{B} = 0$. The two equilibria are therefore limiting cases of a continuous spectrum from which a particular equilibrium is determined by the value of $b_g$. Figure 1a and b show graphical illustrations of Harris and force-free equilibrium current sheets, respectively. A combination of the two profiles yields a mixed equilibrium, as shown in Fig. 1c.

We now argue that a particular equilibrium from the spectrum is attained in the following relaxation process, as illustrated in Fig. 1d. Consider an initially disequilibrated collisionless current sheet whose plasma thermal pressure at the center is lower than the magnetic pressure at the outskirts and which is under a small but finite, uniform seed guide field. As the current sheet pinches to achieve equilibrium, the frozen-in guide field is amplified at the center. The final state is a mixture of a Harris current sheet (with a small uniform guide field) and a force-free current sheet, i.e., mixed equilibrium.

Through this relaxation process, any particular equilibrium can be attained given the appropriate initial conditions, that is, the seed guide field strength and the initial difference between the thermal and magnetic pressures. The details of this process, however, are still unclear. For example, the final state in Fig. 1d exhibits a finite $\partial B_z/\partial x$ that does not exist in the initial state; this corresponds to the development of a finite $J_y$ shear, whose origins are puzzling at this point. This macroscopic alteration must originate from microscopic, single-particle transitions. Therefore, in order to understand the equilibrium selection process, single-particle dynamics in a current sheet with a finite guide field will now be examined.

### Particle orbit classes
For the magnetic field profile $\mathbf{B} = B_0[\hat{y}\tanh(x/\lambda) + \hat{z}b_g]$, one may choose the vector potential $\mathbf{A} = B_0[\hat{y}b_g x - \hat{z}\lambda \ln\cosh(x/\lambda)]$. Because the system is symmetric in the $y$ and $z$ directions, the canonical momenta $p_y = mv_y + qb_gB_0x$ and $p_z = mv_z - q\lambda B_0 \ln\cosh(x/\lambda)$ are conserved quantities, where $m$, $q$, and **v** are the particle mass, charge, and velocity, respectively. The normalized total energy of the particle $\bar{H} = H/\left(m\lambda^2\omega_c^2\right)$ where $\omega_c = qB_0/m$ is then $\bar{H} = \bar{v}_x^2/2 + \chi(\bar{x})$, where

$$\chi(\bar{x}) = \frac{1}{2}\left(\bar{p}_y - b_g\bar{x}\right)^2 + \frac{1}{2}\left(\bar{p}_z + \ln\cosh\bar{x}\right)^2 \quad (4)$$

is the normalized effective potential of the particle motion in the $x$ direction. Here, the barred quantities are normalized, i.e., $\bar{x} = x/\lambda$, $\bar{\mathbf{v}} = \mathbf{v}/(\lambda\omega_c)$, and $\bar{\mathbf{p}} = \mathbf{p}/(m\lambda\omega_c)$.

When $b_g = 0$, i.e., there is no guide field, $\chi$ can have either a single-well shape (for $\bar{p}_z \geq 0$) or a double-well shape (for $\bar{p}_z < 0$). Particle orbits can then be classified into four distinct classes, and current sheet relaxation can be characterized by transitions among the orbit classes[29]. When $b_g$ is finite, however, $\chi$ is asymmetric and is necessarily single-welled when $\bar{p}_z \geq -b_g^2$, but the exact condition for double-welled $\chi$ becomes more complicated (see Methods). Nonetheless, particle orbits can still be classified into three distinct classes, as shown in Fig. 2. Figure 2a–c show examples of $\chi(\bar{x})$ for the three classes. Figure 2d–f show the motion of three particles (red, blue, and cyan) in the $(\bar{x},\bar{z})$ plane for each orbit class; the kinetic energy of each particle is shown as dashed lines with matching colors in Fig. 2a–c. Figure 2g–i are the same as Fig. 2d–f except that they are in the $(\bar{x},\bar{y})$ plane. The orbits were determined using the Boris algorithm[47] under $\mathbf{B} = B_0\left(\hat{y}\tanh\bar{x} + 0.15\hat{z}\right)$.

If $\chi$ is double-welled, a particle may be in one of two orbit classes. The first one is the non-crossing (NC) class, where the

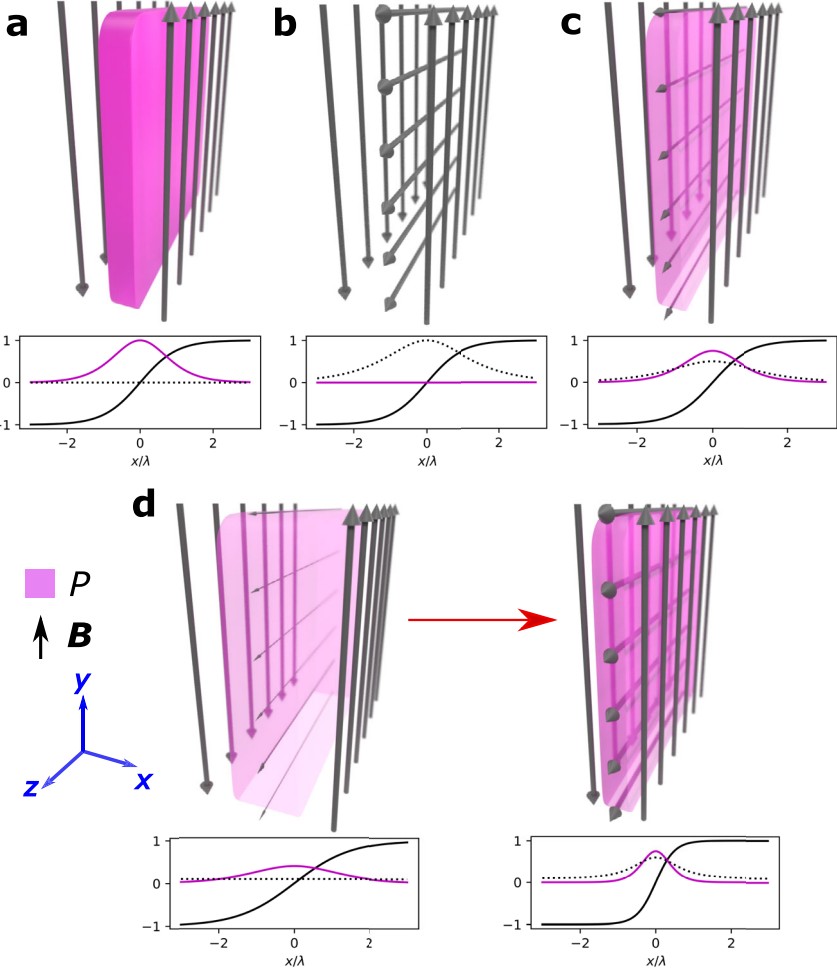

**Fig. 1 | Current sheet types.** Graphical illustrations of the **a** Harris equilibrium, **b** force-free equilibrium, and **c** mixed equilibrium. **d** The proposed process in which a disequilibrated Harris current sheet with a small seed guide field relaxes and locally amplifies the guide field in the process, thereby creating a mixed equilibrium. The arrow directions and widths represent the magnetic field direction and strength, respectively, and the magenta color intensity represents the plasma pressure strength. The line plots are profiles of the shear magnetic field $B_y(x)/B_0$ (black solid line), plasma pressure $P(x)/(B_0^2/2\mu_0)$ (magenta line), and the guide magnetic field $B_z(x)/B_0$ (black dotted line) for each sheet.

particle is trapped in one of the two wells and does not cross $\bar{x} = 0$ (Fig. 2a, d, and g). The second one is the double-well (DW) class, where the particle has enough energy to undergo a full double-well motion (Fig. 2b, e, and h). The blue particle in Fig. 2b belongs to NC, but is plotted to show the NC → DW transition as the particle gains energy. If $\chi$ is single-welled, a particle is in the single-well (SW) orbit class (Fig. 2c, f, and i). For the DW and SW classes, it can be seen that a particle's average $\bar{v}_z$ or $\langle \bar{v}_z \rangle$ changes from negative to positive as the particle gains energy.

Our classification of particle orbits in the finite $b_g$ case is different from that in the $b_g = 0$ case[29] in the following aspects. First, the meandering class in the $b_g = 0$ case is redefined here as the single-well class, because a single-well $\chi$ no longer guarantees meandering (i.e., repeatedly crossing the origin) if $b_g$ is finite. Second, DW is not divided further into two subclasses based on their $\langle \bar{v}_z \rangle$. This is because for $b_g = 0$, only DW particles may have either $\langle \bar{v}_z \rangle > 0$ or $\langle \bar{v}_z \rangle < 0$ since $\langle \bar{v}_z \rangle > 0$ for all SW particles and $\langle \bar{v}_z \rangle < 0$ for all NC particles. For finite $b_g$, however, both DW and SW classes may have either positive or negative $\langle \bar{v}_z \rangle$, so a further division of the DW class is not warranted. Third, $\chi$ is asymmetric and so is particle motion in the $\bar{x}$ direction. Fourth, in contrast to the $b_g = 0$ case, $\bar{v}_y$ is not a constant of motion anymore and changes in time for finite $b_g$; in other words, the distributions of $\bar{v}_y$ are not the same for each class.

## Phase-space distributions

Let us now examine how each orbit class is represented in phase space. $8 \times 10^7$ particles were randomly sampled from the Harris distribution,

$$f(\bar{x}, \bar{\mathbf{v}}) \propto \exp\left(\frac{\bar{v}_x^2 + \bar{v}_y^2 + \left[\bar{v}_z - \bar{V}\right]^2}{2\bar{v}_T^2}\right) \frac{1}{\cosh^2 \bar{x}}, \quad (5)$$

where $\bar{V}$ and $\bar{v}_T$ are the mean drift and thermal velocities normalized by $\lambda\omega_c$, respectively. $\bar{V} = 0.005$ and $\bar{v}_T = 0.05$ were chosen, which satisfy the $\bar{V}_\sigma = 2\bar{v}_{T\sigma}^2$ condition for equilibrium.

Figure 3a–c show particle distributions in $(\bar{x}, \bar{v}_x)$, $(\bar{x}, \bar{v}_y)$, and $(\bar{x}, \bar{v}_z)$ phase spaces, respectively, where each orbit class is represented by one of three—black, blue, and red—colors. Figure 3d–f show the current density components $\bar{J}_i = \int \bar{v}_i f_i d\bar{v}_i$ for $i = x, y, z$, respectively, where $f_i$ is normalized so that $\int\int f_i d\bar{v}_i d\bar{x} = 1$. Figure 3g–i show the diagonal elements of the temperature tensor $\bar{T}_{ii} = \int (\bar{v}_i - \langle \bar{v}_i \rangle)^2 f_i d\bar{v}_i / \int f_i d\bar{v}_i$ for $i = x, y, z$, respectively.

Regarding current sheet equilibration, equilibration of an underheated current sheet requires sheet pinching and heating, i.e., increase of $J_z$ and ion/electron temperatures $T_{i,e}$. For $b_g = 0$, these are achieved by NC → DW transitions induced by particle energization[29], because the DW class has a higher temperature and a higher $\langle \bar{v}_z \rangle$ in comparison to

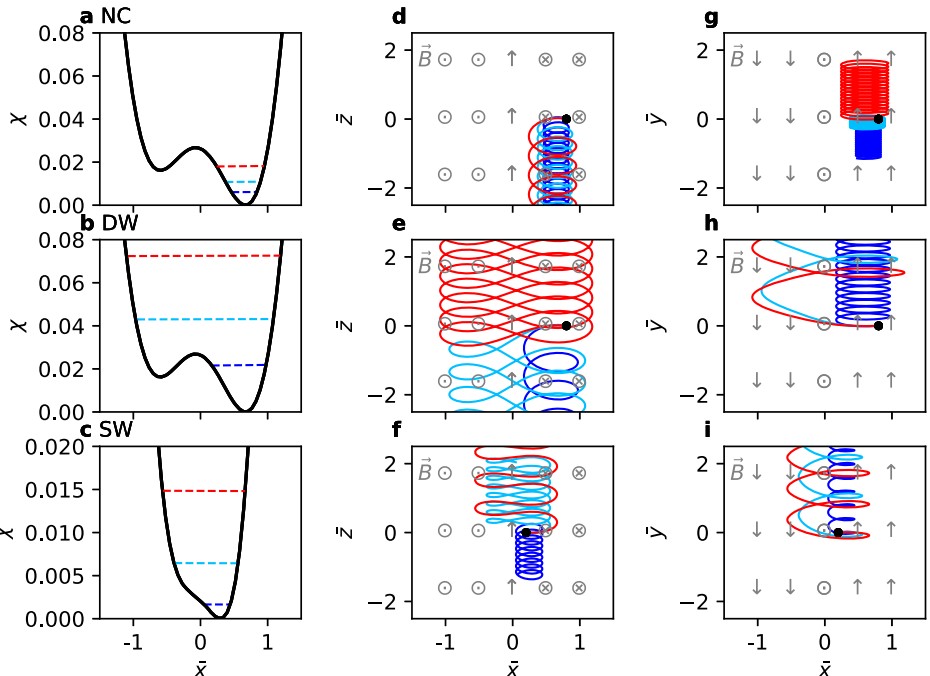

**Fig. 2 | Three classes of particle orbits and their effective potentials.** Effective potentials $\chi$ of the **a** non-crossing (NC) orbit class, **b** double-well (DW) orbit class, **c** and single-well (SW) orbit class. The blue particle in **b** belongs to NC but is plotted to show the NC → DW transition. **d–f** Particle orbits in the $\bar{x} - \bar{z}$ plane respectively belonging to the three classes in **a–c**. Local, approximate directions of the magnetic field are represented by dots, crosses, and arrows. The exact magnetic field profile is $\mathbf{B} = B_0 \left( \hat{y} \tanh \bar{x} + 0.15 \hat{z} \right)$. Three particles are plotted for each class and are labeled by the blue, cyan, and red colors. Each particle's energy is represented by its corresponding color in **a–c**. **g–i** Particle orbits in the $\bar{x} - \bar{y}$ plane.

NC. The heating occurs because the NC → DW transition necessarily involves a passage through the unstable equilibrium as in Fig. 2b, which in turn involves a breakdown of adiabatic invariance and phase-mixing[48]. The pinching occurs due to the sign reversal of $\langle \bar{v}_z \rangle$. For finite $b_g$, sheet equilibration is expected to be similarly achieved by orbit class transitions, but it will be shown later that NC → DW transitions of ions are responsible for heating and NC → SW transitions of electrons are responsible for pinching. As a sidenote, the DW and SW classes may be further divided based on the sign of $\langle \bar{v}_z \rangle$ into two subclasses each (DW ± and SW ± ). The phase space distributions of each class and their respective contributions to density, temperature, and current density are shown in Fig. S1 in Supplementary Materials.

Regarding equilibrium selection, the anti-symmetry in Fig. 3b is important in the transformation from the Harris current sheet with a seed guide field to a mixed current sheet. Recall that our proposed mechanism for the transformation involves the amplification of the seed $B_z$, which is equivalent to the development of a $J_y$ shear. Of the three classes, only the SW class provides the correct shear direction that amplifies $B_z$ because $B_z \sim -\int \bar{J}_y d\bar{x}$ by Ampère's law. For the SW class, $\bar{J}_y \sim \bar{x}$ near $\bar{x} = 0$, so $B_z \sim c_0 - \bar{x}^2$ where $c_0$ is a constant, which is a center-peaked profile. The other classes have $\bar{J}_y \sim -\bar{x}$ near $\bar{x} = 0$, so $B_z \sim c_0 + \bar{x}^2$ which is a center-dipped profile. Therefore, the main current-carrying species must mainly undergo transitions to the SW class.

Combining these two observations gives conjectural answers to the aforesaid two main questions. An initially disequilibrated Harris-type current sheet under a small finite guide field equilibrates through NC → DW & SW transitions. Ions undergo NC → DW transitions, being mainly responsible for heating, and electrons undergo NC → SW transitions, being responsible for pinching. The latter transitions also induce $J_y$ shear, i.e., a current sheet shear, which corresponds to a local increase of $B_z$, leading to a locally amplified guide field and a mixture of Harris and force-free equilibria.

## Particle-in-cell verification

In order to verify the conjectural answers, one-dimensional particle-in-cell simulation were conducted. The initial condition was an under-heated Harris current sheet with temperatures $T_i = T_e = 0.2 T_{eq}$ where $T_{eq} = B_0^2 / (4 \mu_0 n_0 k_B)$ is the Harris equilibrium temperature. The sheet was initially immersed in a uniform seed guide field with strength $b_g = 0.15$. The initial sheet thickness was $\lambda = 10 d_i$ where $d_i$ is the collisionless ion skin depth, and the mass ratio $m_i / m_e = 100$.

A simulation run with $c/v_A = 2$ where $v_A = B_0 / \sqrt{\mu_0 n_0 m_i}$ (Fig. 4) will mainly be referred to in this paper for clarity of presentation; another run with $c/v_A = 20$ was conducted (Fig. S2 in Supplementary Materials), which exhibits plasma oscillations[49–52] that propagate away without affecting the core relaxation mechanism that is orbit class transitions. The current sheet pinches (Fig. 4d) and heats up (Fig. 4e, f) until $t = 30 \omega_{ci}^{-1}$ after which it achieves equilibrium and remains steady. The resultant $J_z$ is not bifurcated, in contrast to the $b_g = 0$ case[29]. The sheet also develops a finite $J_y$ shear (Fig. 4c) which strongly amplifies $B_z$ at the center up to $0.4 B_0$ (Fig. 4b).

Examining the time evolution of ion and electron distribution functions reveals the origin of current sheet heating, pinching, and shearing. It will be shown that ions and electrons undergo different orbit class transitions in the presence of a guide field, making ions mainly responsible for heating and electrons mainly for pinching and shearing.

Figure 5 shows the time evolution of the ion distribution function in $(\bar{x}, \bar{v}_x)$ (a–c), $(\bar{x}, \bar{v}_y)$ (e–g), and $(\bar{x}, \bar{v}_z)$ (m–o) spaces and of the ion current densities $J_{iy}$ (i–k) and $J_{iz}$ (q–s). Here, $f_i(x, v_x) = \iint f_i(x, \mathbf{v}) dv_y dv_z$ and similarly for $f_i(x, v_y)$ and $f_i(x, v_z)$. Figure 5d, h, and p show the difference between the initial and the final states, so the red (blue) color indicates the region to (from) which ions migrate during equilibration. In respective comparisons to Fig. 3a–c, it can clearly be seen that ions mostly undergo NC → DW transitions. Two obvious features indicating this fact are (i) the pronounced slope

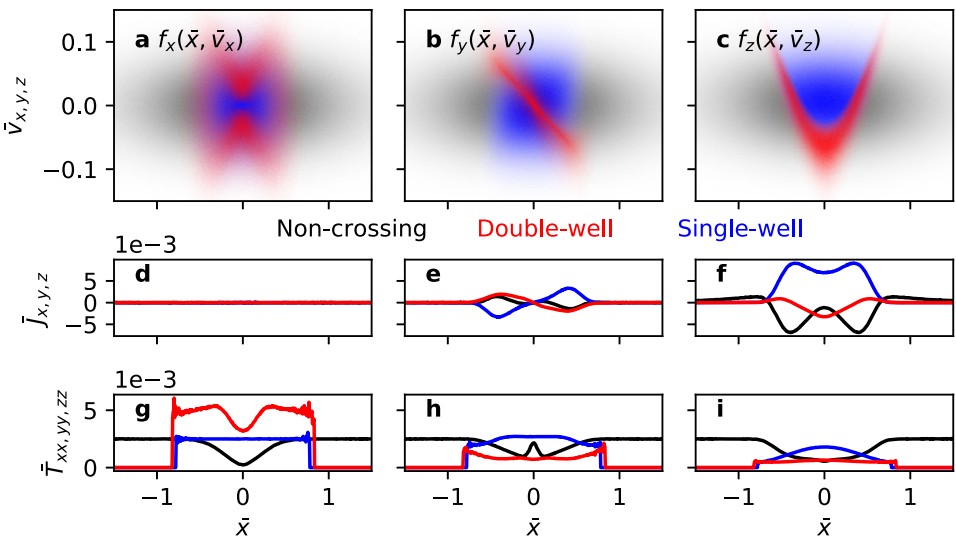

**Fig. 3 | Particle distributions in phase space and their moments.** Phase-space distributions of the three orbit classes distinguished by the black (non-crossing), red (double-well), and blue (single-well) colors in **a** $(\bar{x},\bar{v}_x)$, **b** $(\bar{x},\bar{v}_y)$, and **c** $(\bar{x},\bar{v}_z)$ spaces for uniform $b_g = 0.15$. Current density components **d** $\bar{J}_x$, **e** $\bar{J}_y$, and **f** $\bar{J}_z$, and diagonal components of the temperature tensor **g** $\bar{T}_{xx}$, **h** $\bar{T}_{yy}$, and **i** $\bar{T}_{zz}$ of the three orbit classes.

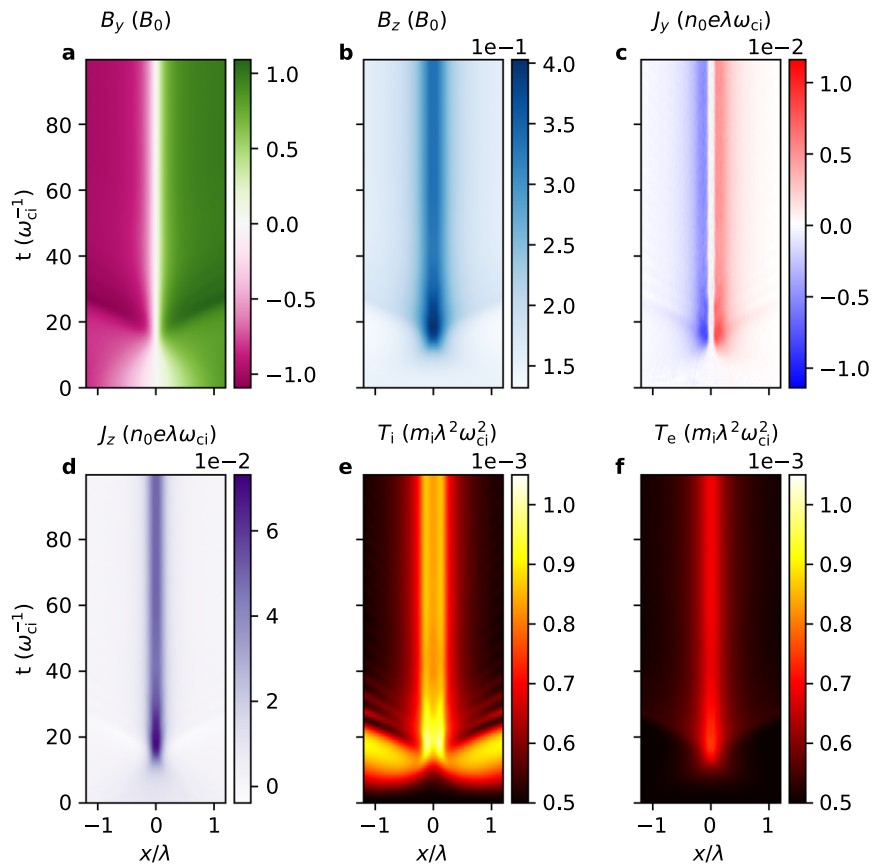

**Fig. 4 | Streak plots of variables from the particle-in-cell simulation.** Streak plots of the magnetic fields **a** $B_y$ (sheared field), **b** $B_z$ (out-of-plane field), the current densities **c** $J_y$ (sheared current), **d** $J_z$ (out-of-plane current), **e** ion temperature $T_i$, and **f** electron temperature $T_e$ from $t = 0$ to $100\omega_{ci}^{-1}$.

that goes like $\bar{v}_y \sim -\bar{x}$ in Fig. 5g that produces $\bar{J}_{iy} \sim -\bar{x}$ near $\bar{x} = 0$ (Fig. 5k), and (ii) the pronounced V-shape in Fig. 5o and the resultant bifurcated $\bar{J}_{iz}$. Ions are heated as a result of this transition because of the aforesaid breakdown of adiabatic invariance and phase-mixing. Therefore, ions are responsible for sheet heating during equilibration, much more so than electrons (compare Fig. 4e to 4f) because

electrons do not mainly undergo NC → DW transitions, as we will see shortly.

Figure 6 is the same as Fig. 5, except that it pertains to electrons. It can be seen from the pronounced void in Fig. 6g and h crossing $(\bar{x},\bar{v}_y) = (0,0)$—corresponding to the slope that is present in Fig. 5g and h—that electrons do not transition to the DW class, but to the SW class.

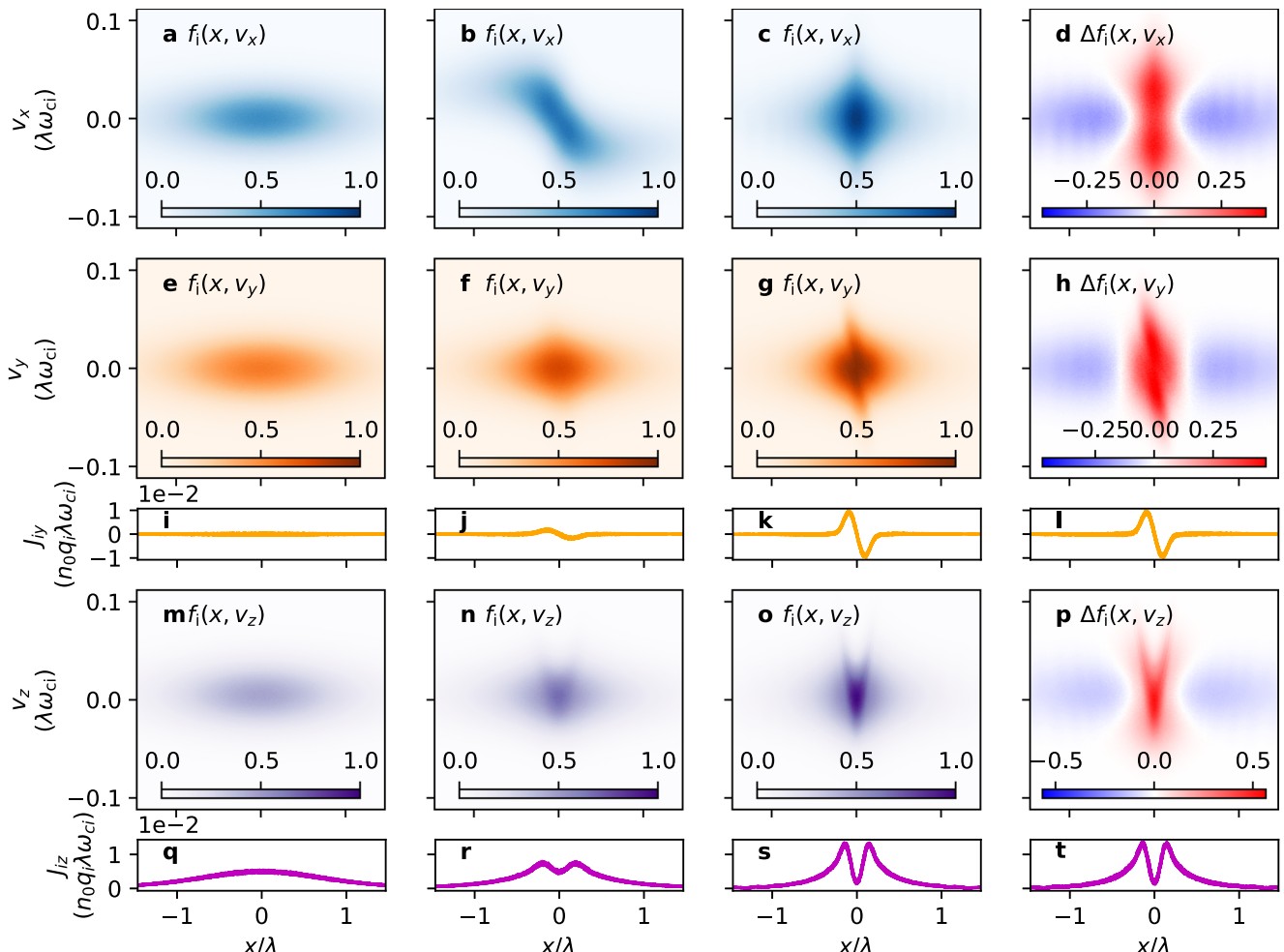

**Fig. 5 | Time evolution of $f_i$ from the particle-in-cell simulation.** Ion distribution function $f_i$ in $(\bar{x}, \bar{v}_x)$ space at **a** $t = 0$, **b** $t = 10\omega_{ci}^{-1}$, and **c** $t = 100\omega_{ci}^{-1}$. **d** The difference ($\Delta f_i$) between $f_i$ in **c** and **a**. **e–h** [**m–p**] are respectively the same as **a–d**, except in $(\bar{x}, \bar{v}_y)$ [$(\bar{x}, \bar{v}_z)$] spaces. In each row, the color represents $f_i$ normalized by the maximum value in the third column (**c**, **g**, and **o**). **i–l** [**q–t**] Ion current density $J_{iy}$ [$J_{iz}$] obtained by taking the first velocity moment of **e–h** [**m–p**].

The development of $\bar{J}_{ey} \sim \bar{x}$ near $\bar{x} = 0$ (Fig. 6k) also indicates this fact. In particular, the enhanced localized current density in Fig. 6s indicates that electrons are responsible for sheet pinching. Electrons are the main current carriers, because the $J_{ez}$ peak in Fig. 6s is about five times larger than the $J_{iz}$ peaks in Fig. 5s (note that $J_z/n_0 q_e \lambda \omega_{ce} = m_e/m_i \times J_z/ n_0 q_i \lambda \omega_{ci}$ and $m_i/m_e = 100$ here). One can also see from Fig. 6s that the electrons transition in such a way that the current density is less bifurcated, as opposed to the $b_g = 0$ case.

NC → SW transitions correspond to a local amplification of the seed guide field near $\bar{x} = 0$ (cf. Fig. 3e). Although the ions' NC → DW transitions induce $J_y$ of the opposite polarity (Fig. 5k), the magnitude of $J_{iy}$ is only half of that of $J_{ey}$. Therefore, the electrons' NC → SW transitions are responsible for sheet shearing and guide field amplification.

The difference in the transition types of ions and electrons arises because the single-well condition ($\bar{p}_{z\sigma} = \bar{v}_{z\sigma}|_{\bar{x}=0} > -b_g^2$) is more sensitive to changes in $b_g$ for electrons than for ions. For $T_i = T_e$, the order of magnitude of $\bar{v}_{zi}$ is much larger than that of $\bar{v}_{ze}$ (compare the scale of Fig. 5o to that of Fig. 6o), and so as $b_g$ increases due to current sheet shearing, the condition $\bar{v}_{ze}|_{\bar{x}=0} > -b_g^2$ is more easily satisfied. This is not the case for $b_g = 0$, so both ions and electrons mainly undergo NC → DW transitions[29] in contrast to the finite $b_g$ case presented here.

## Comparison to spacecraft observations

In order to show the universality of mixed equilibria, we now compare the final equilibrium obtained in the PIC simulation at $t = 100\omega_{ci}^{-1}$ to a reconnecting current sheet measured by the MMS spacecraft from 05:26:41.7 to 05:26:43.7 UT on 3 July 2017, when it was located at $(-17.6, 3.3, 1.7)R_E$—where $R_E$ is the Earth's radius—in Geocentric Solar Magnetospheric (GSM) coordinates while crossing the magnetotail plasma sheet. This current sheet was undergoing magnetic reconnection, whose details were examined in[53].

Figure 7 shows the side-by-side comparison of various quantities between the measured sheet (Fig. 7a–f) and the simulated sheet (Fig. 7g–l). The data are presented in $LMN$ coordinates, where $L$ is the sheared magnetic field direction, $M$ is the guide field direction, and $N$ is the direction normal to the current sheet, i.e., $LMN$ corresponds to $yzx$. The measurement time window translates to 2 s $\simeq$ 200 km[53], which, using the measured electron density $n_e = 0.45$cm$^{-3}$, translates to - 25$d_e$. In the simulation, $\lambda = 10d_i = 100d_e$, so the window in Fig. 7g–l is $0.5\lambda = 50d_e$. The measured electron diffusion region was estimated to be - 6$d_e$, and the corresponding region in the simulation is - 12$d_e$. Also, $b_g \simeq 0.4$ for both sheets. Therefore, the two sheets are similar in that their length scales $L$ satisfy $d_e \ll L \lesssim d_i$, so their dynamics are mainly controlled by electrons, and in that the guide field strengths are the same.

The dashed lines in Fig. 7g and l show analytical solutions that are slightly modified from Eqs. (1) and (2). Because, in the simulation, a seed guide field of strength 0.15 exists and the current sheet pinches down to a length scale $\lambda' \simeq 0.2\lambda$, $B_M = 0.15 + 0.2/\cosh(x/\lambda')$ is plotted (purple dashed line in Fig. 7g). The analytical pressure tensor

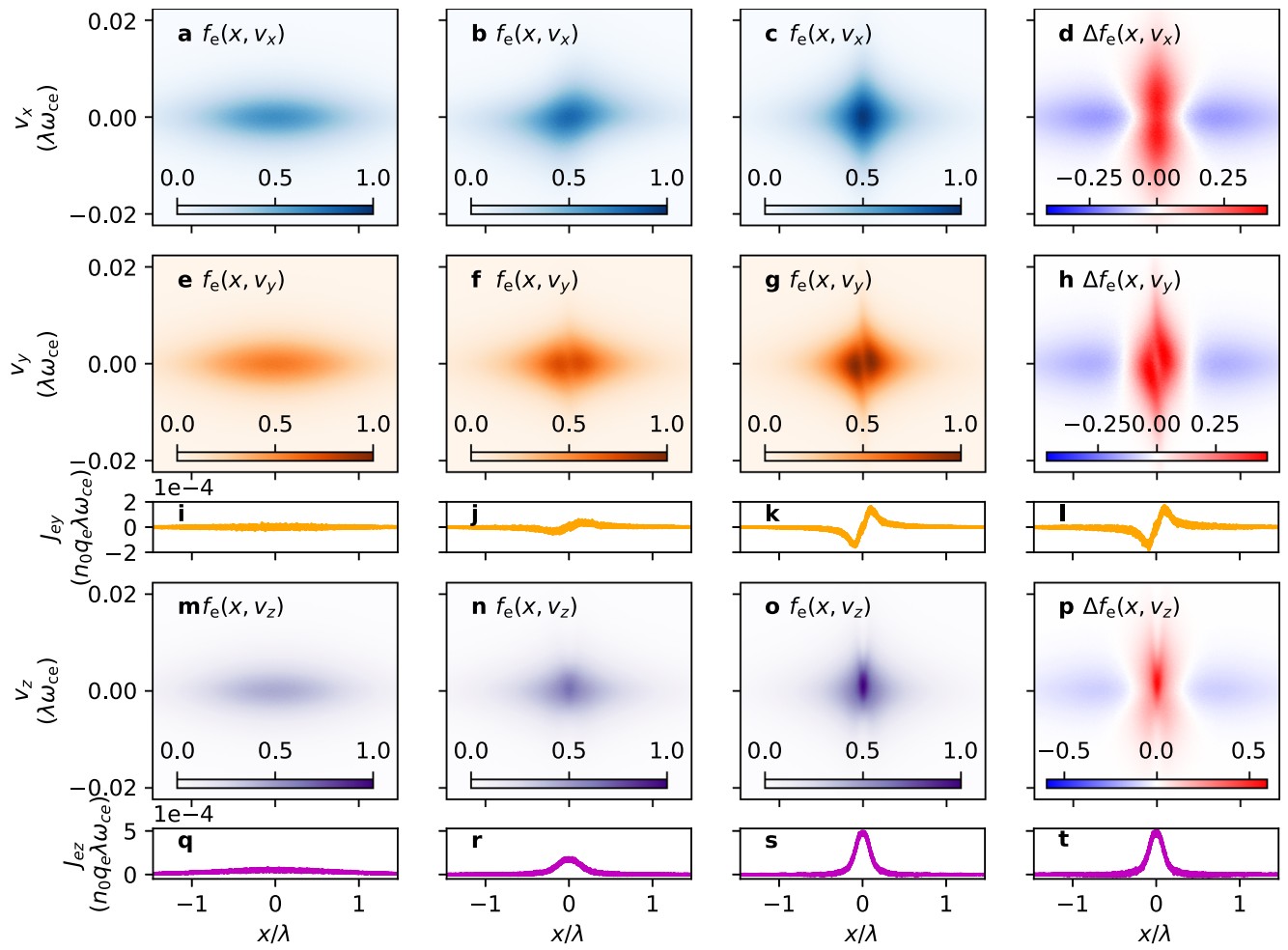

**Fig. 6 | Time evolution of $f_e$ from the particle-in-cell simulation.** Same as Fig. 5, but for electrons.

component $P_{xx}$ was re-calculated accordingly and it was also assumed that $P_{xx}$ is distributed equally among all three components, and therefore $P_{xx}/3$ is plotted (cyan dashed line in Fig. 7l). The agreement between the analytical and numerical solution is evident, save some details in the pressure tensor; strictly speaking, this means that even Eqs. (1) and (2) do not capture the whole spectrum and detailed phase-space analyses are necessary. In fact, there is no reason to expect the initial sheet to relax exactly to one of the equilibria predicted by Eqs. (1) and (2) because even they are merely specific solutions with a number of assumptions (e.g. zero electrostatic potential). The simulated sheet, on the other hand, represents the exact solution attained by the system's path in variable ($P$, **B**, etc) space.

One obvious discrepancy is the central $v_{eL}$ reversal (yellow arrows in Fig. 7b) in the measured sheet that is not present in the simulated sheet (Fig. 7h). This is because of the localized dip in $B_M$ (purple arrow in Fig. 7a; note that $-v_{eL} \sim J_{eL} \sim -\partial B_M/\partial N$), which likely exists due to an initially non-uniform seed guide field with said dip. A PIC simulation run with a center-dipped initial guide field was conducted, and its results reproduce the central $v_{eL}$ reversal (Fig. S3h in Supplementary Materials). Another discrepancy is the rather flat nature of the measured $T_{eLL}$ (Fig. 7d) compared to the simulated $T_{eLL}$ (Fig. 7j). However, this can be explained by the fact that reconnection induces an increase of $T_{eLL}$ at the outskirts[29,54], rendering the profile flat.

Aside from these particulars, the profiles of the measured and simulated sheets agree very well. The global reversal of $v_{eL}$ in Fig. 7h is well reproduced in Fig. 7b, corresponding to the overall center-peaked $B_M$ in Fig. 7a and g. A noteworthy feature is the presence of a double-

peak $T_{eMM}$ (arrows in Fig. 7d) and a single-peak $T_{eNN}$ profile, which is reproduced in Fig. 7j. The agreement is much clearer in Fig. 7e and k, where the parallel temperature is double-peaked and the perpendicular temperature is single-peaked in both sheets. Note that the relative amplitudes between $T_\perp$ and $T_\parallel$ are different in the two cases because the observed sheet and the simulated sheet have different background plasma temperatures. The simulated background temperature starts off isotropic, whereas the observed background temperature is already $T_\parallel > T_\perp$ far away from the sheet for presently unknown reasons. From the agreement between the two sheets, one can see that the measured current sheet arose from an initially disequilibrated state.

## Discussion

Because the measured sheet is reconnecting, it is not in such static equilibrium as the simulated sheet. However, it is evident from our results that many features of 1D current sheet equilibration apply to 2D reconnecting sheets as well. It is therefore important to distinguish effects due to current sheet thinning from those due to reconnection itself. For instance, while the global $v_{eL}$ reversal in Fig. 7b would most likely be interpreted according to the status quo as electron outflows during reconnection, it can also be interpreted as the result of current sheet relaxation leading to guide field amplification as in the present study. Considering that a popular method of identifying reconnection sites is by identifying jet reversals, the ambiguity of the origin of such reversals has important implications.

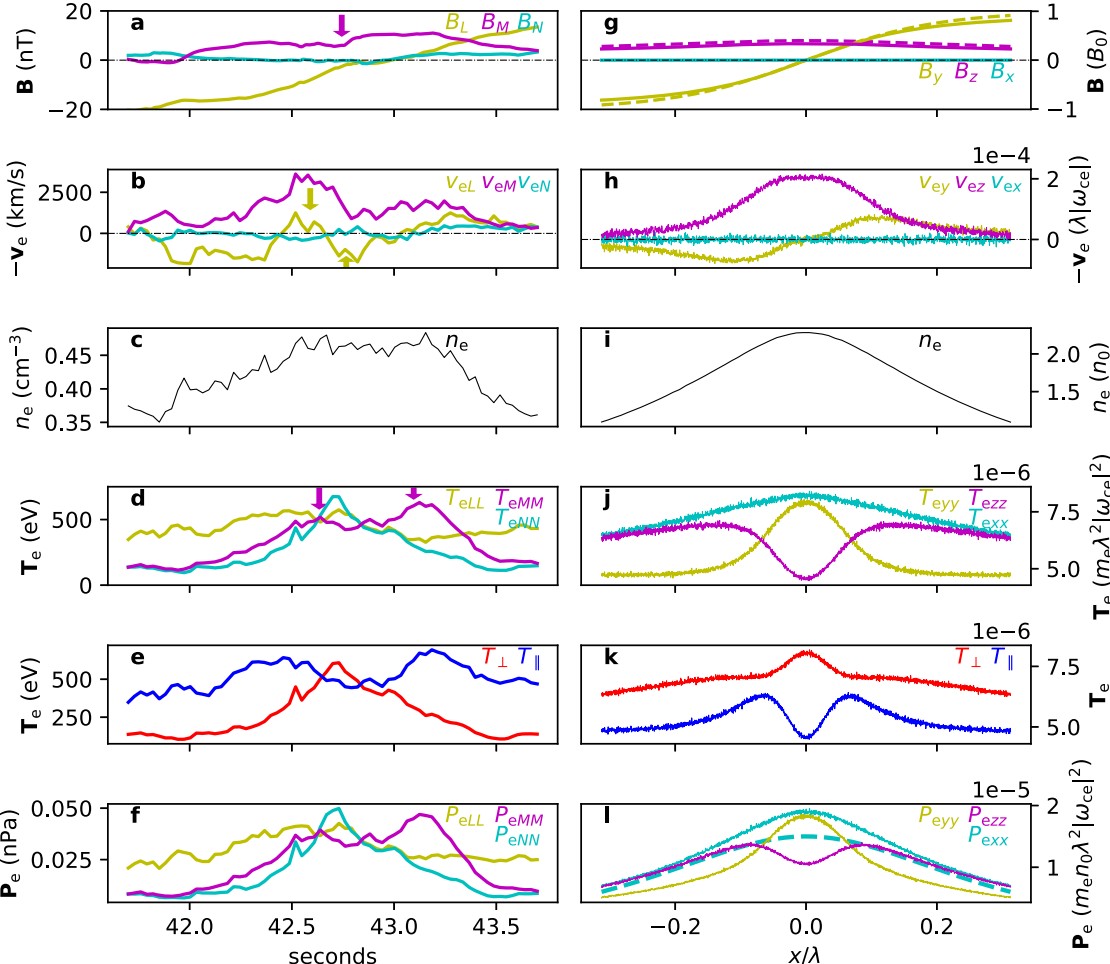

**Fig. 7 | Comparison of MMS data to the PIC simulation. a–f** Sequentially, the magnetic field **B**, electron velocity $\mathbf{v}_e$ (note that the sign is reversed), electron density $n_e$, diagonal elements of the electron temperature tensor $\mathbf{T}_e$, the parallel and perpendicular electron temperatures $T_{e\parallel}$ and $T_{e\perp}$, and diagonal elements of the electron pressure tensor $\mathbf{P}_e$ detected by the Magnetospheric Multiscale spacecraft from 05:26:41.7 to 05:26:43.7 UT on 3 July 2017. The $x$-axis is seconds from 05:26:00 UT, and 1 s corresponds to about $12.5d_e$. **g–l** Quantities from the particle-in-cell simulation respectively corresponding to **a–f**. The $x$-axis is in units of $\lambda = 10d_i = 100d_e$. The dashed lines in **g** and **l** are analytically calculated profiles from Eqs. (1) and (2).

The relaxation time scale is an important factor in determining whether mixed equilibria are ubiquitous, because if the time scale is too long, then other faster phenomena become dominant before equilibration. One obvious indicator is the time it takes for the sheet to completely equilibrate. For the simulation run with $c/v_A = 2$, it takes $\sim 30\omega_{ci}^{-1}$ (Fig. 4), and for the run with $c/v_A = 20$, it takes $\sim 200\omega_{ci}^{-1}$ to settle to a new equilibrium (Fig. S2 in Supplementary Materials). Another indicator is the inductive electric field $E_z$, which is equal to the rate of change of magnetic flux per unit length. For $c/v_A = 2$, it was found that the peak value of $E_z/v_A B_0 \simeq 0.32$ (Fig. S4a in Supplementary Materials), and for $c/v_A = 20$, $E_z/v_A B_0 \simeq 0.6$ (Fig. S4b in Supplementary Materials). Note that there is no $E_z$ after the sheet equilibrates because $\partial \mathbf{B}/\partial t = 0$ and that there is no parallel electric field at the center, so its peak value along flux loops at the outskirts is noted. For comparison, it is widely known that, during collisionless Hall reconnection, $E_z/v_A B_0 \simeq 0.15$ and that it takes around tens to hundreds of $\omega_{ci}^{-1}$ for $E_z$ to reach this value depending on the initial sheet and plasma parameters[25,55]. Therefore, 1D current sheet relaxation is at least as fast as fast reconnection, and so mixed equilibria are expected to be ubiquitous underlying local structures in terms of variations across the current sheet.

An interesting trend in recent current sheet statistics in both the near-Sun and near-Earth solar wind[45,46] is that, the thinner the current sheet is, (i) the stronger its peak current normalized by the Alfvén current $J_A = n_0 e v_A$, (ii) the smaller the current sheet amplitude (amount of magnetic shear relative to the average global magnetic field strength), and (iii) the smaller the shear angle. All these relations are congruous with the proposed equilibration process. As an initially disequilibrated current sheet pinches down to smaller scales, (i) the normalized current becomes stronger, and (ii) the local guide field amplification is stronger so that both the relative current sheet amplitude and the shear angle get smaller. The equilibration process thus explains very well the origin and universality of said statistics.

To further demonstrate universality, we have conducted an additional simulation where the initial temperature is $T = T_{eq}$, not $T = 0.2T_{eq}$, and the density is uniform with $0.2n_0$. The sheet is thus initially disequilibrated due to insufficient density. The macroscopic results—current pinching, shearing, heating, and guide field amplification—are qualitatively similar to Fig. 4 (see Fig. S5 in Supplementary Materials), and the underlying orbit class transitions are similar. Therefore, the equilibration process is insensitive to the initial conditions.

The rapidity of sheet equilibration begs the question of how mixed equilibria change the 2D/3D reconnection dynamics. As mentioned before, reconnection simulations have almost exclusively used as initial conditions the Harris equilibrium (with or without a uniform guide field) and the force-free equilibrium, but mixed equilibria are more likely to be the underlying current sheet in many situations. Also,

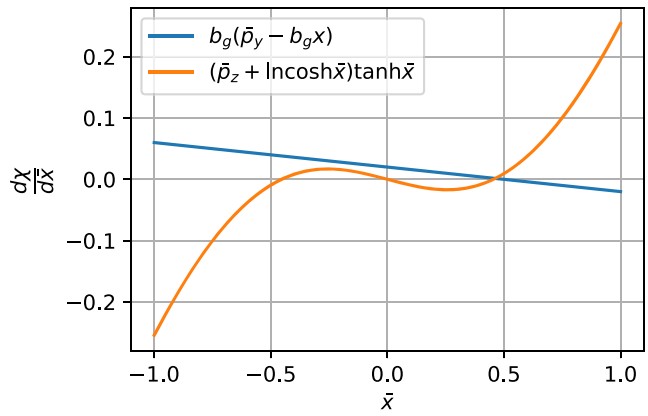

**Fig. 8 | Two functions that contribute to $d\chi/d\bar{x}$ for $\bar{p}_y = 0.1$, $\bar{p}_z = -0.1$, and $b_g = 0.2$.** Plots of functions $f_1(x) = b_g(\bar{p}_y - b_g x)$ and $f_2(x) = (\bar{p}_z + \ln\cosh x)\tanh x$.

preliminary 2D simulations of a large-scale disequilibrated current sheet indicate that, after it rapidly equilibrates down to kinetic scales, reconnection proceeds spontaneously in various places within the sheet without any external perturbations. This is related to the multiple scale problem and/or the onset problem, both of which are important challenges for reconnection research[56].

In summary, the relaxation process of an initially disequilibrated current sheet under finite guide fields was studied. The process involves orbit class transitions similar to the zero-guide-field case, but their details and effects significantly differ. In particular, the transitions reveal the process in which a given plasma system chooses one equilibrium from the spectrum represented by mixtures of the Harris and force-free equilibria. The process was verified by PIC simulations through phase-space analysis and the resultant equilibrium was compared to MMS observations, demonstrating solid mutual agreement and thus the universality of mixed equilibria. Important implications for magnetic reconnection analyses were also discussed.

## Methods
### Particle Orbit Classification
The condition for double-well motion must first be derived. Differentiating Eq. 4 to find its extrema yields

$$\frac{d\chi}{d\bar{x}} = -b_g\left(\bar{p}_y - b_g\bar{x}\right) + (\bar{p}_z + \ln\cosh\bar{x})\tanh\bar{x}. \quad (6)$$

The solutions to $d\chi/d\bar{x} = 0$ are the intercepts between the functions $f_1(\bar{x}) = b_g\left(\bar{p}_y - b_g\bar{x}\right)$ and $f_2(\bar{x}) = (\bar{p}_z + \ln\cosh\bar{x})\tanh\bar{x}$. In order for $\chi$ to be double-welled, there must be three intercepts, one for a local maximum and two for local minima.

Figure 8 shows an example of the two functions. It can be seen that for there to be three intercepts, two conditions must be true. (i) The slope of $f_1$ at $\bar{x} = 0$, which is $\bar{p}_z$, must be steeper than the slope of $f_2$, which is $-b_g^2$. This yields the first condition, which is

$$\bar{p}_z < -b_g^2. \quad (7)$$

(ii) At the points where the slope of the two functions are the same, the absolute value of the linear function must be less than that of the nonlinear function at those points. The condition for the slope of $f_2$ to be $-b_g^2$ is

$$(\bar{p}_z + \ln\cosh\bar{x}) = 1 - \left(1 + b_g^2\right)\cosh^2\bar{x}, \quad (8)$$

which cannot be solved in closed form. Instead, we can make approximations for small $\bar{x}$, $\ln\cosh\bar{x} \simeq \bar{x}^2/2$ and $\cosh^2\bar{x} \simeq 1 + \bar{x}^2$.

Equation (8) then becomes

$$\bar{p}_z + \frac{\bar{x}^2}{2} = 1 - \left(1 + b_g^2\right)\left(1 + \bar{x}^2\right), \quad (9)$$

For which the solutions are

$$\bar{x}_\pm = \pm\sqrt{\frac{-b_g^2 - \bar{p}_z}{3/2 + b_g^2}}. \quad (10)$$

Condition (ii) now becomes simultaneously satisfying $f_1(\bar{x}_+) > f_2(\bar{x}_+)$ and $f_1(\bar{x}_-) < f_2(\bar{x}_-)$, which is for small $\bar{x}_\pm$,

$$b_g\left(\bar{p}_y - b_g\bar{x}_\pm\right) \gtrless \bar{p}_z\bar{x}_\pm. \quad (11)$$

Inserting Eq. (10) and solving yields the condition for $\chi$ to be double-welled:

$$\bar{p}_y^2 < \frac{\left(-b_g^2 - \bar{p}_z\right)^3}{b_g^2\left(3/2 + b_g^2\right)}. \quad (12)$$

Note that Eq. (7) is a necessary condition for there to be a solution for $\bar{p}_y$ in Eq. (12). If a particle does not satisfy Eq. (12), it belongs to the SW class.

Even if $\chi$ is double-welled, a particle still needs to have enough energy to overcome the local hill near $\bar{x} = 0$ in order to undergo full double-well motion. Approximating $f_2$ as $\bar{p}_z\bar{x}$, the location of the local hill is at $\bar{x}_{lh} \simeq b_g\bar{p}_y/\left(\bar{p}_z + b_g^2\right)$. The value of the effective potential at this point is, approximately,

$$\chi(\bar{x}_{lh}) \simeq \frac{1}{2}\bar{p}_y^2\left(1 - \frac{b_g^2}{\bar{p}_z + b_g^2}\right) + \frac{1}{2}\bar{p}_z^2, \quad (13)$$

where terms up to $O(\bar{x}_{lh}^2)$ are kept. If the particle energy is bigger than $\chi(\bar{x}_{lh})$, it belongs to the DW class, and if not, it belongs to the NC class.

Finally, the sign of $\langle\bar{v}_z\rangle$ must be determined if one wants to further classify the DW and SW classes. The bounce-period-averaged $\bar{v}_{z\sigma}$ is given by

$$\langle\bar{v}_z\rangle = \frac{2}{T_0}\int_{\bar{x}_{min}}^{\bar{x}_{max}} \frac{\bar{v}_z}{\bar{v}_x}d\bar{x}, \quad (14)$$

$$= \frac{2}{T_0}\int_{\bar{x}_{min}}^{\bar{x}_{max}} \frac{\bar{p}_z + \ln\cosh\bar{x}}{\sqrt{2H - (\bar{p}_z + \ln\cosh\bar{x})^2 - \left(\bar{p}_y - b_g\bar{x}\right)^2}}d\bar{x}, \quad (15)$$

where $T_0 = 2\int_{\bar{x}_{min}}^{\bar{x}_{max}} d\bar{x}/\bar{v}_x$ is the bounce period, and $\bar{x}_{min}$ and $\bar{x}_{max}$ are solutions to

$$\frac{1}{2}(\bar{p}_z + \ln\cosh\bar{x})^2 + \frac{1}{2}\left(\bar{p}_y - b_g\bar{x}\right)^2 = 0. \quad (16)$$

If $\langle\bar{v}_z\rangle > 0$, the particle belongs to the + class, and to the − class otherwise.

If the DW and SW classes are further divided into DW± and SW± classes based on the sign of $\langle\bar{v}_z\rangle$, there are a total of five orbit classes for a current sheet under a finite guide field. Figure S1 in Supplementary Materials shows the distribution classification of the five orbit classes. Figure S1a–c should be compared with Fig. 3a–c.

## Particle-in-cell simulation

The open-source, fully-relativistic particle-in-cell code, SMILEI[57], was used. The simulation domain was $10\lambda = 100d_i$, which was divided into $2^{15} = 32,768$ grid points. The boundary conditions for the electromagnetic fields and particles were Silver-Müller and periodic, respectively. Two simulations were conducted, one with (i) $c/v_A = 2$ (Fig. 4) and another with (ii) $c/v_A = 20$ (Fig. S2 in Supplementary Materials). For case (i), 10,000 particles were placed per cell and the simulation was run for $100\omega_{ci}^{-1}$ with timestep $\Delta t = 7.63 \times 10^{-4}\omega_{ci}^{-1}$, and for case (ii) 100 particles were placed per cell and the simulation was run for $250\omega_{ci}^{-1}$ with timestep $\Delta t = 7.63 \times 10^{-5}\omega_{ci}^{-1}$. In both cases, the initial condition was a Harris sheet with the ion and electron temperatures set as one-fifth of the equilibrium temperature, i.e., $T_{i,e} = 0.2T_{eq}$ where $T_{eq} = B_0^2/(4\mu_0 n_0 k_B)$, and with a uniform seed guide field of strength $0.15B_0$. Streak plots of the out-of-plane inductive electric field $E_z$ are shown for (i) $c/v_A = 2$ (Fig. S4a in Supplementary Materials) and (ii) $c/v_A = 20$ (Fig. S4b). The peak values of $E_z$ are around 0.3 and 0.6, respectively.

The simulations were run on the KAIROS computer cluster at Korea Institute of Fusion Energy.

## MMS data and LMN coordinate system

MMS1, MMS2, and MMS3 data from 05:26:41.7 to 05:26:43.7 UT on 3 July 2017 were averaged to yield the profiles in Fig. 7a–f. MMS4 data were omitted because it was not available from the import method provided by the pySPEDAS package[58], which was used for extracting the data. The magnetic field data and plasma data were collected by the Fluxgate Magnetometer instrument[59] and the Fast Plasma Investigation instrument[60], respectively. The local LMN coordinate system obtained in Chen et al.[53] was used, and the unit vectors are, in Geocentric Solar Magnetospheric coordinates, $L = (-0.9840, 0.1685, 0.0578)$, $M = (0.1781, 0.9360, 0.3036)$, and $N = (0.0030, -0.3090, 0.9511)$.

## Guide field dip

Another particle-in-cell simulation was conducted where an initial dip was applied in the guide field, i.e., the initial $B_z$ was $B_z = b_g(1 - 0.5/\cosh(x/\lambda)^2)$. Figure S3h successfully reproduces the observed local reversal of $v_{eL}$ in Fig. S3b. To reduce the computational cost, this simulation was run with 4,096 grid points and 100 particle-per-cell, so the noise level is much higher than that in Fig. 7.

## Data availability

MMS data are publicly available from https://lasp.colorado.edu/mms/sdc/public. The fiducial PIC simulation data generated in this study have been deposited in the Zenodo database (https://doi.org/10.5281/zenodo.6395310).

## Code availability

SMILEI[57] is an open-source particle-in-cell code available from https://smileipic.github.io/Smilei. MMS data were analyzed using the pySPEDAS package[58], available from https://github.com/spedas/pyspedas. The codes used in the data analyses are available from Y.D.Y. upon reasonable request.

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

## Acknowledgements

This work was supported by the National Research Foundation (NRF) of Korea under grant Nos. NRF-2019M1A7A1A03088456 (G.S.Y.), NRF-2022M3I9A1073808 (Y.D.Y.), and NRF-2019R1A2C1004862 (Y.D.Y.), and by the Ministry of Science and ICT under grant No. 2021-SRETC-P01-2 (Y.D.Y.). D.E.W. was supported by the NASA Magnetospheric Multi-Scale Mission in association with NASA contract NNG04EB99C. Y.D.Y. was supported by the PIURI fellowship of POSTECH, the POSCO Science Fellowship of POSCO TJ Park Foundation, and by an appointment to the JRG Program at the APCTP through the Science and Technology Promotion Fund and Lottery Fund of the Korean Government. Y.D.Y. was also supported by the Korean Local Governments—Gyeongsangbuk-do Province and Pohang City. The particle-in-cell simulations presented here were conducted on the KAIROS computing cluster at Korea Institute of Fusion Energy.

## Author contributions

Y.D.Y. performed the theoretical analysis, performed the simulations, analyzed the spacecraft data, and wrote the manuscript. D.E.W. contributed to the interpretation of the simulation and observation results, as well as to the revision of the draft. G.S.Y. conceived the central idea, oversaw the project, and provided general guidance.

## Competing interests

The authors declare no competing interests.
