## [Transparent Peer Review File · Nature Communications]

Equilibrium selection via current sheet relaxation and guide field amplificationREVIEWER COMMENTS

Reviewer #1

This paper presents a possible process by which to determine the equilibrium outcome of the relaxation of a disequibrated current sheet in the presence of a small finite guide field. This is an extension of their previous work of the same process without a guide field being present.

It is my opinion that this work will provide a basis and framework for future work in current sheet behavior.

Although it need not be included in this work, additional study should be conducted with higher current sheet aligned energies and with higher mass species.

I feel that this paper should be published after addressing of several minor comments.

General Comment.

Too many unneeded modifiers. These words obfuscate the message of the paper. This makes it incredibly more difficult to read. Words like "are exhaustively classified" should be "are classified". "in order to fathom the equilibrium" should be "in order to understand the equilibrium" fathom is more than a little "over the top"

Specific Comments.

Was this study run in 2D or 3D? I assume 2D since you made "out of plane" streak plots, but this is not explicitly stated.

FIG 1 is not clear enough to make sense. Consider more traditional line plots of B and rho as cuts in the YX plane. The arrow widths are nearly impossible to decipher and the pink on pink color choice was just bad. What does it mean when the lines are closer or further apart in d)? Explain this to the reader if you maintain these plots.

In the text body below Figure 1 you explain "Fig 1d exhibits a finite dBz/dx that does not exist in the initial state" How can we see dBz/dx when the only thing shown is B not Bz (which I must assume is total magnetic field strength)? If you do not expect to show dBz/dx in FIG. 1d then you should not refer to it, or refer to the appropriate place.

In FIG. 2, how were the ion orbit trajectories determined? Was this done in 2D or 3D?

FIG 2 Do the red (cyan, blue) lines match between column 2 and column 3? How have you defined meandering?

How does a meandering (M) orbit differ from the other two? Should there really only be only two kinds of orbits, DW and NC?

Why does meandering in this paper appear to have a different meaning than in your previous paper [29]?

FIG 2 is confusing. Figures a-b-c make sense. you specified that X is normal to the CS.

Am I correct in assuming that:

- 1) Z direction is along the CS.
- 2) Y is across the CS.

3) that your XYZ would be ZXY in GSE coordinates (tail).

As such Column 2 (d-e-f) shows movement along the CS and Column 3 (g-h-i) across the CS? Am I correct in assuming that particle motion is along the CS (by some means) and across the CS due to the Guide Field?

Particles can move in both + and - directions along the current sheet and across the current sheet.

In FIG 2 what decides whether the red, blue or cyan will move: one way or the other (+/-Z) along the CS (d-e-f)?

In FIG 2 what decides whether the red, blue or cyan will move: one way or the other (+/-Y) across the CS (g-h-i)?

I see that later you specify subclasses based on the sign of

-- Figure 2 The green and magenta colors represent By. I assume the colors are plus and minus but it is not stated. --

I retract this initial comment and replace it with:

I suggest that you move the By and Bz arrows above the column or give them a white background. Pink on pink and Green on green is very hard to see.

What is your XYZ coordinate system? It does not seem to be any standard system such as GSE.

Also you are comparing XYZ with LMN which is given in reference to GSE. This produces confusion.

Please clarify how the LMN coordinate system relates to your XYZ system? I mean in general terms like \sim along the CSTM, \sim perpendicular to the CSTM, \sim across the CSTM.

FIG 4 FIG 5 the color bars could be a little bit wider. They are difficult to see.

FIG 4 the parentheses get very confusing when you have embedded ((x,yz)). I understand your desire to try and duplicate statements for two rows that are slightly different.

Try using e-h {m-p} are respectively the same as a-d, except in (x,vy)

{(x,vz)} $\hat{=}$ or similarly [square brackets]. $\hat{=}$ [q-t] Ion current density Jiy [Jiz] obtained by taking the first velocity moment of e-h [m- p]. $\hat{=}$

Reviewer #2 (Remarks to the Author: Overall significance):

This paper concerns the problem of relaxation of an out of equilibrium current sheet in collision-less plasmas. The authors are interested in describing how the new equilibrium is selected. They present a theoretical derivation of particle orbits, particle-in-cell numerical simulations, and observations. I don't have any major concerns about the quality of the work or how the paper is currently written.

Reviewer #2 (Remarks to the Author: Impact):

The subject of the paper is of great importance and potential impact.

Reviewer #2 (Remarks to the Author: Strength of the claims):

The claim of "universality" that the author makes doesn't seem to be supported enough by the analysis the authors present. In other words, the authors claim that the process they have studied is universal, but the paper does not contain convincing proof of such a statement. For this reason, although the scientific quality of the work is excellent, I don't think that its current version meets the journal requirements.

In what follows, I list those points that, in my opinion, should be improved to arrive at a more convincing version of this study.

1. As it is commonly done in PIC simulations, the authors use a nonrealistic speed of light to Alfvén speed

ratio (c/v_a). The results presented in the Extended Data section show that a larger value of c/v_a results in the production of waves. The authors state that these plasma oscillations damp away without affecting the core relaxation mechanisms. I have a few questions on this point. Which kind of waves are these? Why do they develop when a larger c/v_a is considered? What damps these waves? What is the effect of periodicity on the evolution of these waves inside the system? In an open system, would such waves leave the current sheets? Are these waves observed by MMS or any other satellite?

2. The simulations the authors have performed are 1D. However, the stability of a single current sheet may depend on 3D dynamics. What justifies the 1D approach the authors use? In their conclusions, the authors also admit that in a 2D case they have studied, reconnection spontaneously develops, which makes questionable the existence in natural environments of a 1D equilibrium like what they are presenting in the paper.

3. Could the authors better explain why the perpendicular electron temperature in their simulation is everywhere larger than the parallel one (Fig 6 panel e-f) while it is the opposite in the observed current sheet? They make an argument on that just before the section "Discussion and outlook"; however, it was hard for me to understand what they mean.

4. The authors show that their numerical simulations are in accordance with their analysis of particle trajectories. Would it be possible for them to analyze MMS data and extract the VDF for ions and electrons? How do these compare with those obtained from numerical simulations and the analysis of particle orbits?

5. At the beginning of page 14, the authors note a discrepancy between the analytical and the numerical solution. This discrepancy seems to be a weak point of the paper since the numerical solution does not relax to one of the expected equilibria. Still, it relaxes to some other equilibrium whose analytical form is unknown. It would be a great result if the authors could find an analytical form corresponding to the final state of their simulations.

6. In their final discussion, the authors claim that mixed equilibria are likely to be ubiquitous. However, they also admit that their MMS observations concern a current sheet out of equilibrium, and this seems an apparent contradiction. Are the type of equilibria discussed by the authors also frequently observed by MMS?

Minor comment.

In Fig 4 and Fig 5 a few distribution functions of the kind $f(x,v)$ are plotted. Could the authors specify how the two not plotted velocities are treated? For instance, when v_x is plotted, do they plot a particular plane at fixed v_y and v_z or the average on v_y and v_z ?

Reviewer #2 (Remarks to the Author: Reproducibility):

I don't see any significant issue in reproducing the analysis presented in the paper.

We deeply thank both reviewers for their supportive and constructive comments. We addressed their concerns in the revised manuscript and believe that the revisions have greatly improved the manuscript.

Reviewer #1

This paper presents a possible process by which to determine the equilibrium outcome of the relaxation of a disequilibrated current sheet in the presence of a small finite guide field. This is an extension of their previous work of the same process without a guide field being present. It is my opinion that this work will provide a basis and framework for future work in current sheet behavior. Although it need not be included in this work, additional study should be conducted with higher current sheet aligned energies and with higher mass species. I feel that this paper should be published after addressing of several minor comments.

We thank the reviewer again for the supportive comments. Your feedback has particularly improved the legibility and clarity of our manuscript. The additional studies that are mentioned will be looked at in future works.

General Comment.

Too many unneeded modifiers. These words obfuscate the message of the paper. This makes it incredibly more difficult to read. Words like “are exhaustively classified” should be “are classified”. “in order to fathom the equilibrium” should be “in order to understand the equilibrium”
fathom is more than a little ‘over the top’

We apologize for the illegibility. Those and several other modifiers were taken out to be clearer and more concise.

Specific Comments.

Was this study run in 2D or 3D? I assume 2D since you made “out of plane” streak plots, but this is not explicitly stated. FIG 1 is not clear enough to make sense. Consider more traditional line plots of B and ρ as cuts in the YX plane. The arrow widths are nearly impossible to decipher and the pink on pink color choice was just bad. What does it mean when the lines are closer or further apart in d)? Explain this to the reader if you maintain these plots.

In the text body below Figure 1 you explain “Fig 1d exhibits a finite dBz/dx that does not exist in the initial state” How can we see dBz/dx when the only thing shown is B not Bz (which I must assume is total magnetic field strength)? If you do not expect to show dBz/dx in FIG. 1d then you should not refer to it, or refer to the appropriate place.

The current sheet structure in question is 1D in configuration space and 3D in velocity space, and we have specified this information after Equation 3. Regarding Figure 1, we fully agree with your comments and have made adjustments that incorporate your input. After trying different options including just presenting 2D plots, changing colors, etc, we think that the best option is to present the line plots alongside the original plots. This removes all ambiguity and illustrates the exact mathematical profiles while maintaining the intuitive picture of the original plots. Figure 1’s caption was also appropriately modified.

The issue with dBz/dx is also resolved with the line plots.

In FIG. 2, how were the ion orbit trajectories determined? Was this done in 2D or 3D? FIG 2 Do the red (cyan, blue) lines match between column 2 and column 3?

The trajectories were determined in 3D using the standard Boris algorithm under the given magnetic field profiles. We specified this in line 143.

The colors indeed match among columns 2 and 3. This is in fact already specified both in the caption (“Three particles ... corresponding color in a-c.”) and in the text (lines 139-142), which we believe are sufficient.

How have you defined meandering?

How does a meandering (M) orbit differ from the other two?

Should there really only be only two kinds of orbits, DW and NC?

Why does meandering in this paper appear to have a different meaning than in your previous paper [29]?

Thanks to your questions, we have re-evaluated our definitions. Originally, we had defined meandering as particles that have single-well effective potentials. This was fine in the zero-guide-field case because a single-well potential necessarily meant that the motion is “meandering,” i.e., repeatedly crossing the origin. We mindlessly extended the usage to the guide field case, but as your confusion points out, single-well particles no longer necessarily undergo meandering orbits (as can be seen in Fig. 2f).

Therefore, we have changed the M class to the single-well (SW) class and specified this in line 141. The nomenclature should now be free of confusion because the orbits are clearly defined with respect to the effective potential of each particle. To reiterate, a single-well particle is in the SW class. A double-welled particle is in the DW class if it has enough energy to overcome the local maximum, or in the NC class if it doesn't and is trapped in one of the wells.

As you mention later and as we have already demonstrated in the paper, the classes can be subdivided into many subclasses based on the directions of the particles' average velocities in the y and z directions. However, the important question is: what is the minimum number of class divisions necessary to understand the current sheet relaxation and guide field amplification process? We believe the current number of divisions (3) is the minimum.

FIG 2 is confusing. Figures a-b-c make sense.

you specified that X is normal to the CS.

Am I correct in assuming that:

- 1) Z direction is along the CS.
- 2) Y is across the CS.
- 3) that your XYZ would be ZXY in GSE coordinates (tail).

As such Column 2 (d-e-f) shows movement along the CS and Column 3 (g-h-i) across the CS? Am I correct in assuming that particle motion is along the CS (by some means) and across the CS due to the Guide Field?

Regarding the coordinate system, you are correct that our XYZ is ZXY in GSM (not GSE; was a typographical error and is now corrected). We seriously considered changing our coordinate system to more align with GSM, but we decided to not change it, mainly because of the current paper's relationship with Yoon et al., 2021 (Ref 29). Readers referring to Yoon2021 will be extremely confused if we change the coordinate system.

In any case, the important comparison in the current paper is between xyz and LMN. We extended our definition of xyz after Equation 3, and specified in line 266 that LMN corresponds to yzx.

Particles can move in both + and - directions along the current sheet and across the current sheet.

In FIG 2 what decides whether the red, blue or cyan will move: one way or the other (+/-Z) along the CS (d-e-f)?

In FIG 2 what decides whether the red, blue or cyan will move: one way or the other (+/-Y) across the CS (g-h-i)?

I see that later you specify subclasses based on the sign of

The particles velocities in the y and z directions must be orbit-averaged in order to figure out whether they move in + or – directions (for example, see Eq. S10 in Methods). As we mentioned above, we can of course subdivide the classes, but the number of classes should be kept to a minimum to avoid conceptual cluttering when understanding the core point of the paper. In any case, the velocity mean/spread of each class is well demonstrated in the phase-space distributions.

-- Figure 2 The green and magenta colors represent B_y . I assume the colors are plus and minus but it is not stated. –

I retract this initial comment and replace it with:

I suggest that you move the B_y and B_z arrows above the column or give them a white background. Pink on pink and Green on green is very hard to see.

Apologies for the confusion. We changed Figure 2 d-i; the colors were replaced with approximate local directions of the magnetic field. The purpose of Figure 2 is more qualitative than quantitative, so the exact profile of the magnetic fields need not be explicit in the figure. The exact profile is now given in the caption.

What is your XYZ coordinate system? It does not seem to be any standard system such as GSE. Also you are comparing XYZ with LMN which is given in reference to GSE.

This produces confusion.

Please clarify how the LMN coordinate system relates to your XYZ system? I mean in general terms like 'along the CS', 'perpendicular to the CS', 'across the CS'.

Again, apologies for the confusion, and we have addressed this point in response to your former comment. As far as I know, the origin of LMN is that N is "Normal" to the current sheet, L is along the shear magnetic field ("Longitudinal") and M completes the LMN system. We already specified this in line 264-266, but we have now re-specified this for our xyz system and have explicitly compared LMN to yzx. Additionally, we added labels for each direction in Fig. 7g-l so that the comparison is more explicit.

FIG 4 FIG 5 the color bars could be a little bit wider. They are difficult to see.

We have doubled the colorbar widths.

FIG 4 the parentheses get very confusing when you have embedded $((x,yz))$. I understand your desire to try and duplicate statements for two rows that are slightly different.

Try using e-h $\{m-p\}$ are respectively the same as a-d, except in (x,v_y) $\{(x,v_z)\}$ or similarly [square brackets]. \hat{e}_i Ion current density J_{iy} $[J_{iz}]$ obtained by taking the first velocity moment of e-h $[mp]$.

Thank you for your great point. Square brackets are now used for enhanced clarity.

Reviewer #2

This paper concerns the problem of relaxation of an out of equilibrium current sheet in collision-less plasmas. The authors are interested in describing how the new equilibrium is selected. They present a theoretical derivation of particle orbits, particle-in-cell numerical simulations, and observations. I don't have any major concerns about the quality of the work or how the paper is currently written.

The subject of the paper is of great importance and potential impact.

We thank the reviewer again for the supportive comments. We also thank the reviewer for perusing the manuscript and providing useful input on the meaning and impact of the manuscript.

The claim of "universality" that the author makes doesn't seem to be supported enough by the analysis the authors present. In other words, the authors claim that the process they have studied is universal, but the paper does not contain convincing proof of such a statement. For this reason, although the scientific quality of the work is excellent, I don't think that its current version meets the journal requirements.

We interpret a “proof” of universality of our claims to contain the following three elements.

1. Are there many observed cases of such mixed equilibria?

Statistical studies of current sheets in the magnetopause [Panov et al., JGRSP (2011)], Jovian magnetotail [Artemyev et al., PSS (2014)], near-Sun solar wind [Lotekar et al., ApJ (2022)], and near-Earth solar wind [Vasko et al., ApJ (2022)] show that mixed equilibria are a significant majority of the observed current sheets. For example, Artemyev2014 reports that out of the 226 observed current sheets, 64 are Harris type (their type #1), 39 are force-free (their type #3), and 123 are mixed (their type #2). Panov2011 also reports that out of the 52 observed current sheets, half of them have what they call “C-shaped” rotation of the magnetic field, which corresponds to the mixed equilibria in our present manuscript. Lotekar2022 and Vasko2022 examine more than 10,000 current sheets each and conclude that the average current sheet is near-force-free with a slight dip of the magnetic field strength at the center (mixed equilibrium) with significant variation in the profiles.

We added information about these observations in the introduction (lines 65-74)

2. Are the mixed equilibria achieved by the equilibration process we show here?

An interesting statistic provided by Lotekar2022 and Vasko2022 is that in both cases, the thinner the current sheet, (i) the stronger its normalized peak current, (ii) the smaller the current sheet amplitude (amount of magnetic shear relative to the average global magnetic field strength), and (iii) the smaller its shear angle. All these relations are congruous with our relaxation process. As the current sheet pinches down to smaller scales, (i) the normalized current becomes stronger, (ii) the local guide field amplification is stronger, so both the relative current sheet amplitude and the shear angle get smaller. Our equilibration process thus explains very well the origin of these statistics, consolidating its universality.

We added this discussion in lines 334-342.

3. Is the equilibration process qualitatively similar for other disequibrated states?

Instead of an under-hot current sheet, we performed an additional, under-dense simulation where the temperature was set as the equilibrium temperature, but the density was set uniformly as one-fifth of the equilibrium density. The end-result is similar to the under-hot case, and the similar orbit-class transitions take place. This fact supports the universality of our results in that the equilibration process is insensitive to the initial conditions.

We added this discussion in lines 343-348, and another figure in the Supplementary Materials.

In what follows, I list those points that, in my opinion, should be improved to arrive at a more convincing version of this study.

1. As it is commonly done in PIC simulations, the authors use a nonrealistic speed of light to Alfvén speed ratio (c/v_A). The results presented in the Extended Data section show that a larger value of c/v_A results in the production of waves. The author state that these plasma oscillations damp away without affecting the core relaxation mechanisms. I have a few questions on this point. Which kind of waves are these? Why do they develop when a larger c/v_A is considered? What damps these waves? What is the effect of periodicity on the evolution of these waves inside the system? In an open system, would such waves leave the current sheets? Are these waves observed by MMS or any other satellite?

The waves are MHD eigenmodes of the sheet, as meticulously analyzed by G. Fruit, JGR, 107, 1411 (2002) and G. Fruit, JGR, 107, 1412 (2002) in the case of a Harris sheet. Intuitively speaking, the plasma is dense and unmagnetized at the center but sparse and magnetized at the outskirts, so there is a sloshing back and forth between plasma density and the magnetic fields, but the wave also travels outwards (left and right). Because the sheet reaches ion inertial scales, the details of the wave may be a bit different from the MHD treatment, but the overall picture should be the same (like Alfvén vs. inertial Alfvén). The simulation is open-boundary, not periodic, so the waves just propagate away (the particle boundary condition is periodic, but the particle density is nearly zero at the boundaries).

The waves actually do develop for $c/v_A=2$ (see Fig 4c and 4e) but propagate away much faster than the $c/v_A=20$ case; this is obvious because the waves are Alfvén at the outskirts, and so higher v_A means that the waves travel away much faster.

Concerning wave damping (if this is what the referee means by “dumping”), we believe it is mainly outward wave propagation, judging from the decay time. For $c/v_A=2$, we expect the wave to travel $1/\lambda$ in about $t \sim 20/w_{ci}$, and for $c/v_A=20$ in about $t \sim 200/w_{ci}$. This is consistent with the simulation results. If the referee means wave generation, then the source of the wave is the initial over-compression of the plasma

density due to pinching.

Observations of these oscillations by Cluster are reported in Louarn, JGR, 109, A03216 (2004) and Fruit, JGR 109, A03217 (2004).

We have cited the above papers where we mention the waves (line 212)

2. The simulations the authors have performed are 1D. However, the stability of a single current sheet may depend on 3D dynamics. What justifies the 1D approach the authors use? In their conclusions, the authors also admit that in a 2D case they have studied, reconnection spontaneously develops, which makes questionable the existence in natural environments of a 1D equilibrium like what they are presenting in the paper.

We are not saying in any way that the mixed equilibria are expected to be present as is, but that such structures are likely to exist as “underlying local structures” in many phenomena (much like how a current-carrying plasma column is an underlying structure for the kink instability). For example, we present below another comparison between an MMS-observed reconnecting current sheet and a 1D PIC equilibration simulation from one of our previous papers (Yoon et al., 2021; Ref. 29). Because of reconnection, there are certain discrepancies, e.g., the existence of an electron outflow v_{eL} in panel b and high T_{eLL} in panel d, but one can see that the underlying structure is more or less explained by the 1D profiles. We changed line 332 to say “underlying local structures.”

Even in the 2D case we mentioned, the equilibration process happens first and establishes a new underlying structure, and the onset of reconnection proceeds at a later time. There are in fact many cases where 1D equilibrium-like structures are locally observed in dynamic events such as magnetic reconnection, e.g., Harris sheet in the Magnetic Reconnection eXperiment (MRX) in Yamada et al., PoP, 7, 1781 (2000) or Lorentz balance in PIC reconnection simulations in Li et al., PRL, 101, 215001 (2008). Basically, the plasma and electromagnetic profiles in the inflow direction are in large part explained by 1D structures. Another example would be the existence of bifurcated current structures during 2D reconnection in the zero-guide-field case; in Yoon et al. (2021), it was shown that 1D current sheet pinching can explain said bifurcation.

One can also look at this problem from the following perspective. In the present manuscript, we show good agreement between an observation of a reconnecting sheet and a simulation of a 1D sheet. Same goes for similar comparisons in Yoon et al. (2021). Also, as mentioned above, magnetopause, magnetotail, and solar wind current sheet statistics support our equilibration process and mixed equilibria. We believe all these should be taken as “evidence” that 1D processes and structures explain various (but certainly not all) aspects of the observed local profiles, even during 2D/3D events such as reconnection. Therefore, we opine that, instead of asking “why are we examining 1D profiles when natural environments are 2D/3D?”, the question for future work should be “why do 1D profiles explain so well the local structures that are globally in 2D/3D?”

Also, it is apparent from our recent line of work that even 1D processes are not understood very well. From a didactic perspective, it is prudent to fully understand 1D systems and processes first before trying to understand 2D/3D.

3. Could the authors better explain why the perpendicular electron temperature in their simulation is everywhere larger than the parallel one (Fig 6 panel e-f) while it is the opposite in the observed current sheet? They make an argument on that just before the section "Discussion and outlook"; however, it was hard for me to understand what they mean.

We apologize for the confusion. It's just that the background temperature are different in the two cases. We start the simulation with a uniform, isotropic temperature, but in the observation, the background plasma (further away from the current sheet) has $T_{\text{par}} > T_{\text{perp}}$ for presently unknown reasons. It could be due to beam effects, but significant out-of-scope effort is necessary to uncover its origin. We revised lines 302-305 to be clearer.

4. The authors show that their numerical simulations are in accordance with their analysis of particle trajectories. Would it be possible for them to analyze MMS data and extract the VDF for ions and electrons? How do these compare with those obtained from numerical simulations and the analysis of particle orbits?

We fully agree that a direct comparison of the distribution functions would be very interesting and it is on our to-do list. However, we would prefer not to include such comparisons in the present manuscript and leave them aside for future work for the following reasons. (i) MMS/PIC comparison is only one of the main points in this paper, the others being particle orbit classification, phase-space analysis, and equilibration process identification. (ii) In light of the present manuscript's focus on phase-space dynamics, a phase-space distribution comparison would be better than a VDF comparison. However, since the former has seldom been done in the past relative to the latter, we need more time and effort in validating and presenting the comparison. All in all, we cannot perform this task without significant additional work beyond the scope of this paper per se.

5. At the beginning of page 14, the authors note a discrepancy between the analytical and the numerical solution. This discrepancy seems to be a weak point of the paper since the numerical solution does not relax to one of the expected equilibria. Still, it relaxes to some other equilibrium whose analytical form is unknown. It would be a great result if the authors could find an analytical form corresponding to the final state of their simulations.

We actually think the discrepancy is one of the strong points of the paper. Even the analytical solution presented in Eqs. 1-3 represents a set of specific solutions that comes from a number of assumptions (e.g., zero electrostatic potential; see Harrison et al., PRL (2009)). As long as the distribution function is ANY function of the constants of motion, it is a valid stationary solution. Therefore, we have no reason to expect at all that the initial current sheet would relax to Eqs. 1-3; we are just plotting the analytical solutions in Fig. 7 for reference. The discrepancy in fact shows that we really need to think of these equilibria in terms of phase-space distributions of particle orbit class, as we have done in this manuscript. We have added lines 282-286 specifying this fact.

Also, we actually tried deriving an exact analytical formula, but the derivation proved to be unwieldy. It is not possible at the present time.

6. In their final discussion, the authors claim that mixed equilibria are likely to be ubiquitous. However, they also admit that their MMS observations concern a current sheet out of equilibrium, and this seems an apparent contradiction. Are the type of equilibria discussed by the authors also frequently observed by MMS?

We believe our answer to the question of universality, as well as our answer to referee's question number 2, jointly answer this question.

Minor comment.

In Fig 4 and Fig 5 a few distribution functions of the kind $f(x,v)$ are plotted. Could the authors specify how the two not plotted velocities are treated? For instance, when v_x is plotted, do they plot a particular plane at fixed v_y and v_z or the average on v_y and v_z ?

The other variables are integrated. The definitions are now given in lines 224-225.

REVIEWERS' COMMENTS:

Reviewer #1 (Remarks to the Author: Overall significance):

Reviewer #1

Accept without editorial revisions.

This paper presents a possible process by which to determine the equilibrium outcome of the relaxation of a disequilibrated current sheet in the presence of a small finite guide field. This is an extension of their previous work of the same process without a guide field being present. It is my opinion that this work will provide a basis and framework for future work in current sheet behavior. Although it need not be included in this work, additional study should be conducted with higher current sheet aligned energies and with higher mass species. I feel that this paper should be published.

I do not request a response to or opportunity to review the following comment.

You responded to one of my original comments with:

Regarding the coordinate system, you are correct that our XYZ is ZXY in GSM (not GSE; was a typographical error and is now corrected). We seriously considered changing our coordinate system to more align with GSM, but we decided to not change it, mainly because of the current paper's relationship with Yoon et al., 2021 (Ref 29). Readers referring to Yoon2021 will be extremely confused if we change the coordinate system.

In any case, the important comparison in the current paper is between xyz and LMN. We extended our definition of xyz after Equation 3, and specified in line 266 that LMN corresponds to yzx.

You indicate that you seriously (as opposed to frivolously?) considered using GSM but chose to stay with the GSE coordinates of Yoon et al. 2021 to avoid confusion. Unfortunately, your revised paper indicates GSM which is aligned with the data coming from Chen et al. 2019 (July 3, 2017). .

This said, your paper does not refer to MMS data from Yoon et al. 2021 (June 17, 2017) at all. Rather it refers only to Chen et al. 2019. Yoon refers to GSE coordinates while Yoon refers to GSM. GSM is the appropriate reference frame to mention.

I believe only your response is in error.

Reviewer #1 (Remarks to the Author: Impact):

.

Reviewer #1 (Remarks to the Author: Strength of the claims):

.

Reviewer #1 (Remarks to the Author: Reproducibility):

.

Reviewer #3 (Remarks to the Author: Overall significance):

The authors have responded to the posed questions in a satisfactory way.

My primary concern was about the universality of the relaxation process. It is more evident now that the authors mean that mixed equilibria exist as underlying local structures in many physical environments.

I have one remaining minor concern.

I think that the authors should rephrase the last part of their abstract, explaining better what they mean by saying that mixed equilibria are universal.

We sincerely thank the reviewers for the fruitful comments, which have resulted in a greatly improved manuscript.

Reviewer #1

Accept without editorial revisions.

This paper presents a possible process by which to determine the equilibrium outcome of the relaxation of a disequilibrated current sheet in the presence of a small finite guide field. This is an extension of their previous work of the same process without a guide field being present. It is my opinion that this work will provide a basis and framework for future work in current sheet behavior. Although it need not be included in this work, additional study should be conducted with higher current sheet aligned energies and with higher mass species. I feel that this paper should be published.

I do not request a response to or opportunity to review the following comment.

You responded to one of my original comments with:

Regarding the coordinate system, you are correct that our XYZ is ZXY in GSM (not GSE; was a typographical error and is now corrected). We seriously considered changing our coordinate system to more align with GSM, but we decided to not change it, mainly because of the current paper's relationship with Yoon et al., 2021 (Ref 29). Readers referring to Yoon2021 will be extremely confused if we change the coordinate system.

In any case, the important comparison in the current paper is between xyz and LMN. We extended our definition of xyz after Equation 3, and specified in line 266 that LMN corresponds to yzx.

You indicate that you seriously (as opposed to frivolously?) considered using GSM but chose to stay with the GSE coordinates of Yoon et al. 2021 to avoid confusion. Unfortunately, your revised paper indicates GSM which is aligned with the data coming from Chen et al. 2019 (July 3, 2017).

This said, your paper does not refer to MMS data from Yoon et al. 2021 (June 17, 2017) at all. Rather it refers only to Chen et al. 2019. Yoon refers to GSE coordinates while Yoon refers to GSM. GSM is the appropriate reference frame to mention.

I believe only your response is in error.

We apologize for the continued confusion. By "seriously considering changing coordinates," we meant changing from XYZ to ZXY, not from GSE to GSM. The LMN coordinates in GSM in the present manuscript are fine as is, as you point out correctly.

As you correctly stated, this confusion only pertains to my response and the manuscript is free of error.

Reviewer #3 (Remarks to the Author: Overall significance):

The authors have responded to the posed questions in a satisfactory way.

My primary concern was about the universality of the relaxation process. It is more evident now that the authors mean that mixed equilibria exist as underlying local structures in many physical environments.

I have one remaining minor concern.

I think that the authors should rephrase the last part of their abstract, explaining better what they mean by saying that mixed equilibria are universal.

We are very glad that our response was satisfactory. We have rephrased the last part of the abstract to be clearer about what we mean.